# Examining the convergence of dominant themes related to social entrepreneurship, NGOs and globalization–A systematic literature review

**Muhammad Rizwan Hussain[1], György Norbert Szabados[1], Khalid Bin Muhammad[2], Sevinj Omarli[3], Shah Ali Murtaza[4]\*, Edina Molnár[5]**

1 Faculty of Economics and Business Institute of Sports Economics and Management, University of Debrecen, Debrecen, Hungary, 2 College of Computer Science and Information Systems, Institute of Business Management, Karachi, Pakistan, 3 Doctoral School of Management & Business, Corvinus University of Budapest, Budapest, Hungary, 4 Károly Ihrig Doctoral School of Management & Business, University of Debrecen, Debrecen, Hungary, 5 Psychology Department, Social Sciences Institute, Faculty of Health Sciences, University of Debrecen, Debrecen, Hungary

\* shah.ali.murtaza@econ.unideb.hu

**Data Availability Statement:** All relevant data are within the paper and its Supporting Information files.

## Abstract

Social entrepreneurship (SE) is an all-encompassing concept in comparison to a typical non-government organization (NGO). It is a topic that has captured the interest of academics investigating nonprofit, charitable, and nongovernmental organizations. Despite the interest, few studies have examined the overlap and convergence of entrepreneurship and non-governmental organizations (NGOs), in congruence with the new phase of globalization. The study gathered and evaluated 73 peer-reviewed papers using a systematic literature review methodology, mainly from Web of Science but also from Scopus, JSTOR, and Science Direct, and supplemented by a search of existing databases and bibliographies. Based on the findings, 71 percent of studies suggest that organizations must reconsider the concept of social work, which has evolved rapidly, aided by globalization. The concept has changed from the NGO model to a more sustainable one, such as that proposed by SE. However, it is difficult to draw broad generalizations regarding the convergence of context-dependent complex variables such as SE, NGOs, and globalization. The results of the study will significantly contribute to a better understanding of the convergence of SE and NGOs, as well as the recognition that many aspects of NGOs, SE, and post-COVID globalization remain unexamined.

## Introduction

Social entrepreneurship and non-governmental organizations (NGOs) are getting more and more attention because they have the potential to help solve social and environmental problems, contribute to economic and social growth, and complement or add to the ways that social and environmental problems are usually solved. Despite numerous reviews on social

**Funding:** The authors received no specific funding for this work.

**Competing interests:** The authors have declared that no competing interests exist.

entrepreneurship, NGOs, and globalization, there are few studies that have explored all three concepts collectively, implying a gap in existing research. This research is important because it gives an overview of the literature on the always-changing landscape of entrepreneurship. It looks at how entrepreneurship has changed into social entrepreneurship, how non-governmental organizations have changed, and how globalization has affected Social entrepreneurship (SE) and non-government organization (NGO). Economists define entrepreneurship as a mix of innovation and risk-taking. Risk-taking and entrepreneurship are crucial features of management decision-making tasks [1]. When entrepreneurship thrives, it leads to higher growth rates and increased opportunities for all, including the poor and marginalized segments of society. Entrepreneurship is seen as a driving force for the creation of jobs and economic growth, which eventually leads to wealth creation for a nation [2]. Social entrepreneurship is a key concept in the social economy. The social economy is a subset of the national economy that is oriented toward social purposes and employs socially acceptable methods of arranging economic activity [3] Social economy has reemerged as a vital source of employment, economic development, social solidarity, associationalism, and social services [4].

Social economy is important as Governments cannot be depended on entirely for economic growth. This is the reason why the disciplines of social enterprise (SE) and non-governmental organizations (NGOs) are receiving increased attention from academics and researchers. Academics have increasingly highlighted the significance of the non-governmental sector in economy in particular social economy. The relation-ship between entrepreneurship and economic development has been extensively re-searched and shown to be positive [5]. Such research has pushed governments to place a greater emphasis on social entrepreneurship. Social entrepreneurs continue to be major economic development and monetary inclusion drivers for marginalized groups of society, particularly in developing countries.

Social entrepreneurs focus on addressing social challenges and driving change via innovation [6]. They concentrate on mobilizing existing resources to create social systems in response to societal problems. Many people feel that social entrepreneurship is a powerful social catalyst and that social entrepreneurs operate as change agents. They define their purpose and societal ideals, find new opportunities, and participate in the process of innovation via adaptation and learning. They act decisively and boldly, despite limited resources, and adhere to mission-related accountability requirements [7]. One hallmark of social entrepreneurs is their willingness to share their acknowledgement of labor [6]. Social entrepreneurship is a broad concept that asserts that social change may be achieved on a much broader and more significant scale than a traditional non-governmental organization (NGO). The process through which people, companies, and entrepreneurs create and finance solutions that directly address social challenges is known as social entrepreneurship (SE). Social economies are more inclusive of marginalized individuals and segments of society than for-profit enterprises [8]. This paper is one of the few that will look at the major themes in the literature about social entrepreneurship, non-governmental organizations (NGOs), and how globalization affects both. Earlier academics, such as Giddens (1984) and Wright (1982), saw social entrepreneurship as a humanitarian and idealistic endeavor [5]. Social entrepreneurship in developing economies is a new notion that is heavily reliant on the government, not necessarily for money, but rather for supporting it through legislation.

Globalization has a significant influence on the functioning of "nongovernmental organizations." The global effect of digital technology has reinforced the concept of globalization, with far-reaching social, cultural, economic, and political implications for the whole world, including advanced, developing, and impoverished nations [9]. To benefit from a phenomenon like globalization, local businesses will need more than simply a feeling of responsibility toward

social causes; they will need initiative in the form of social entrepreneurship. Corporations with well-defined finances and operational areas play an important role in bringing about change through social entrepreneurs and corporate social responsibility (CSR) programs.

Non-Governmental Organizations (NGOs) are increasingly recognized as important actors in civil society and have become critical to socioeconomic development in developing countries. Researchers Mahfuz et al., (2019) and Barba et al., (2022) have emphasized that NGOs are effective because they focus on specific issues, have the ability to mobilize resources, and can act independently of governments. They also provide funding for poverty reduction and socio-economic development initiatives. As such, their role is evolving and becoming more significant in the social economy. The social economy has been increasingly recognized for its importance in addressing and mitigating the impacts of the COVID-19 crisis. Research has explored the sources of knowledge of economic and social value in the sport industry, as well as the significance of the social dimension and role of ethics and economics. Additionally, studies have examined causes and consequences of income inequality from a global perspective.

This research makes an important contribution to the existing literature by examining the intersection of social entrepreneurship (SE), NGOs, and globalization. Through a review of relevant literature and the use of bibliometric and thematic analysis, the research investigates the main themes of studies on these topics. By doing this, the research hopes to find out more about how SE, NGOs, and globalization relate to each other. There is a growing recognition of the role that SE and NGOs can play in addressing social and environmental challenges, as well as the potential for these organizations to contribute to economic and social development. However, there is a lack of research on the intersection of these topics, making this study an important addition to the literature. The findings of this research will provide valuable insights for practitioners, policymakers, and researchers interested in SE, NGOs, and globalization, and will help to inform future research and practice in these areas. The objectives of the study are as follows:

1. To understand how research on social entrepreneurship (SE), non-governmental organizations (NGOs), and globalization has evolved over time by analyzing the literature in these fields from different historical periods.

2. To identify the most cited and influential authors in the field of social entrepreneurship (SE), non-governmental organizations (NGOs), and globalization by conducting a bibliometric analysis of the literature in these fields.

3. To determine the most influential work on the convergence of social entrepreneurship (SE), non-governmental organizations (NGOs), and globalization by conducting a content analysis of the literature in these fields and identifying key themes, theories, and findings.

4. To examine the trends of cooperation among authors and countries in studies converging on the subjects of social entrepreneurship (SE), non-governmental organizations (NGOs), and globalization by analyzing the co-authorship and geographical distribution of publications in these fields.

5. To identify the common themes and most-referred theories among researchers in the field of studies converging on the issues of social entrepreneurship (SE), non-governmental organizations (NGOs), and globalization by conducting a thematic analysis of the literature in these fields.

The study is significant because it seeks to understand how research on social entrepreneurship (SE), non-governmental organizations (NGOs), and globalization has evolved over time,

to identify the most cited and influential authors, to determine the most influential work, to examine cooperation trends among authors and countries, and to identify common themes and most-referred theories in these fields. This study will provide a comprehensive understanding of the current state of research in these fields, which will be useful for researchers, practitioners, and policymakers interested in the intersection of SE, NGOs, and globalization. It will also aid in identifying gaps in the literature and areas for future research. important because it tackles the critical need to understand the current state of research in the disciplines of social entrepreneurship, nongovernmental organizations, and globalization. In recent years, these disciplines have attracted considerable attention and have significant implications for tackling social and environmental concerns. Understanding how research in these domains has progressed over time, identifying significant authors and works, and analyzing trends in international collaboration will give invaluable insights into the present level of knowledge and aid in directing future research efforts. In addition, this study will give a complete review of these domains and help fill any gaps in the literature, making it a useful resource for academics, practitioners, and policymakers.

## Literature review

Social entrepreneurship is a growing field that is gaining recognition for its potential to create positive social and environmental change through innovative business models. The works of researcher Barbara Sanchez have highlighted the importance of SE in addressing pressing issues and fostering a more sustainable and equitable future. Social entrepreneurship is essential for addressing complex social and environmental challenges that cannot be solved through tradition-al means [10]. In her article in the journal Frontiers in Psychology, Sanchez argues that social entrepreneurship can be a powerful force for change by providing innovative solutions to these challenges and driving progress towards a more sustainable future. Barba et al., (2022) also emphasizes the importance of education and training in fostering a new generation of socially-minded entrepreneurs [10]. In her article Sanchez argues that education programs can play a crucial role in supporting the development of the necessary skills and knowledge for aspiring social entrepreneurs to succeed. To recognize and understand the current themes in SE literature it is important to under-stand how research on SE, NGOs have evolved over time which will be examined via research question 1 (RQ1).

Previously, researchers like Mintzberg (1973), Giddens (1984) and Wright (1982) have paid particular attention to entrepreneurship for decades. Despite the interest in entrepreneurship, studying the impact of globalization on SE and NGOs has been the subject of very few studies. To comprehend SE, it is necessary to first explain the distinction between SE and Business entrepreneurship (BE). Business entrepreneurship (BE) and social entrepreneurship (SE) are two wings of a flying creature. Any feathered creature cannot fly without its wings, similarly, any economy cannot progress by just concentrating on one type of an enterprise and ignoring the other. With the changing landscape of the business world, businesses are pushed to contribute economically as well as socially and environmentally. Business enterprises (BE) and social enterprises (SE) both include some risk-taking, chasing goals that many view as unattainable, developing networks and connections, and seeking unique solutions to problems. BE and SE have similarities in that they both build businesses and provide an environment for their employees to flourish and prosper [11]. Despite the fact that the factors described above are similar, there are significant differences. While social entrepreneurship must contend with bad market circumstances in order to have a positive financial and social effect, business entrepreneurship seeks profitable market conditions. If we want to assess success, BE maximizes profit and stock value. SE, on the other hand, believes in long-term societal transformation.

The commercial entrepreneur seeks to reinvest profits in the business, while the social entrepreneur seeks to share profits with stakeholders. BE rigidly adheres to norms and regulations, while SE promotes teamwork, compassion, and respect. BE seeks to generate economic value, while SE seeks to create both economic and social value. BE seeks a risk-free business environment, while SE creates economic and social value in the face of risk and uncertainty [12]. This paper examines the pattern of cooperation amongst authors to understand the current trends about SE, NGOs and globalization via research question 2 (RQ2).

The authors Mahfuz et al., (2019) emphasize the significance of identifying the various SE characteristics. SE combines traditional entrepreneurship with the mission of "promoting social change through entrepreneurial innovation and resilience" [13, 14], Social entrepreneurship (SE) is an entrepreneurial process with a social purpose that aims to create economic and social value by combining innovative ideas and resources [13]. The SE, unlike an NGO or a CSR, is not a charity since it functions purely on a commercial basis. For example, Jeff Skoll, co-founder of eBay, donated 4.4 million British pounds to establish the "Centre for Social Entrepreneurship Research" [12]. Many enterprises in the field of social entrepreneurship have developed in recent years, with one of the most noteworthy being "Sekem," launched by Ibrahim Abouleish to solve critical issues in Egypt [15, 16]. This effort was recognized as worthy of an alternative Nobel Prize, emphasizing the relevance of similar initiatives. SE cannot be understood merely in economic terms; it must also be considered in terms of social and environmental context. Let's look at several theories on SE to have a greater understanding of it: -

I.  According to Giddens' 1984 Structuration Theory, the actor (the social entrepreneur) cannot be separated from the structure (society) [17]. According to the concept, the structure is both a product and a constraint on human conduct not be separated from the structure (society). According to the concept, the structure is both a product and a constraint on human conduct. The Edhi Foundation in Pakistan is the best example of structuration theory, with the example of Abdul Sattar Edhi (a social entrepreneur) changing the socioeconomic condition of society. This concept is crucial because it provides a fresh viewpoint on how the environment impacts the development of social entrepreneurs and the process of social change.

II. Institutional Entrepreneurship: The purpose of institutional entrepreneurship is to explain how institutions are produced and transformed [18]. This concept emphasizes the relevance of institutions in the creation of entrepreneurial initiatives for social change. It also states that fewer established players are motivating entrepreneurs to take part in social entrepreneurship activities that change the laws and conventions.

III. Social Capital and Social Movement theory, Social change is the aim of social capital and social movement theories. This theory's goal is to explain how social entrepreneurs use an entrepreneurial mindset to identify, manage, and overcome social concerns. According to Reed et al. (2002), the concept has highlighted four essential challenges that SE must solve. First, the political climate was examined, including its risks and opportunities [19]. Strategic management focuses on resource mobilization and resource distribution. Following that, it established collective action plans, which resulted in the formation of a unique identity, supported by problem-solving via innovation and networking.

Literature review conducted by Javed et al. (2019), identifies four dimensions of SE. For starters, the SE's social goal lies between for-profit and non-profit organizations [12, 20]. Second, social entrepreneurs are classified as social innovators who develop new products, services, and technology to address a social need [21, 22]. The third factor is social networks, which are linked groups of individuals and organizations that share information and resources. Businesses are no longer self-sufficient and can no longer prosper on their own. The

fourth issue is how financial returns are managed; SE prioritizes profit reinvestment for social purposes. According to researchers Murphy and Coombes (2009) the difference between social entrepreneurship and profit-oriented businesses is in their aims and objectives [23]. The basic goal of SE is to find long-term solutions to social challenges [24]. They do this by maintaining the organization's financial stability and independence. Social entrepreneurs tackle and resolve problems that governments or companies have overlooked by collaborating and using assets, capacities, services, and technology to produce long-term solutions [25]. They need resources and knowledge in order to exist [26]. Social networks provide SE resources, money, and information [27], help in the construction of relationships with individuals and in society [28], knowledge, bridge gaps between stakeholders, minimize risk, and build trust between parties [28]. SE is seen to be a plausible method for generating both economic and social benefits [29]. A review of the existing literature, however, reveals that there are few studies that examine the impact of globalization on both SE and NGO at the same time, indicating a research gap. This study will attempt to fill such gaps in previous studies.

Non-governmental organizations (NGOs) are organizational actors that are not linked with the government or the commercial sector; they represent communities, social and political movements, and ideas, and they operate at all geographies, from local to global [30]. However, few of the previous studies and theories have gone into depth or accounted for the consequences of globalization. This is significant since no company, including SEs and NGOs, operates in isolation. Local and regional policy, the economy, culture, and technology are all affected by globalization. Globalization has allowed non-governmental organizations (NGOs) to play a more major role in global governance, and they have become increasingly essential players in tackling global challenges [31]. According to the authors, "the development of global networks and the rising interdependence of global players have encouraged the rise and impact of non-governmental organizations, allowing them to play a more major role in global governance." Globalization has enhanced the visibility and effect of non-governmental organizations (NGOs), but it has also introduced new problems, such as the need to compete for financing and recognition in a congested global marketplace [32, 33]. As the globe becomes increasingly inter-connected, non-governmental organizations (NGOs) will continue to play an important role in global governance, and will be well positioned to capitalize on the possibilities and difficulties of globalization to accomplish their objectives [34, 35]. Socially entrepreneurial behavior and, more importantly, social innovation—new approaches to social problems, resourcefulness, a larger scale and wider impact, and transferable, scalable, and cost-effective solutions—are important future directions for all types of non-governmental organizations (NGOs), including those whose primary mission is empowerment and social justice [36].

Through the activities of non-governmental organizations (NGOs), social entrepreneurship (SE) has become a vehicle for satisfying the social needs of the poor [32]. The majority of non-governmental organizations (NGOs) do not function in the traditional non-profit paradigm, which is devoid of political influence and engagement [37]. Researchers are becoming increasingly interested in investigating financially viable enterprises. Financially viable social enterprises are autonomous organizations that aim to provide social benefit while also achieving financial sustainability through trade [38]. In light of the above, it is evident that the connection between NGOs and SE is complex since it is made up of interconnected processes and requires further investigation. This study will analyze this relationship by looking at research questions 3, 4, and 5.

## Materials and methods

The method used in the study is literature review methodology, which has been defined by Fink (2020) as a systematic, explicit, reproducible method used to identify, evaluate, and

interpret the current body of literature written on the topic [39]. To identify the most appropriate search terms, we studied the search strategies of relevant published systematic reviews and explored the indexing systems of selected databases to fetch related subject headings [40]. This paper examined existing literature from books, news articles, websites, journals, research papers, and other relevant material on the topic to further explore and understand the convergence of the concepts of SE, NGOs, and globalization.

The study's methodology is based on the work of (Saunders et al. (2019), who are prominent and recognized researchers in the field of research methodology, providing credibility to the study [41]. The methodology is favored since it provides a systematic and planned strategy to complete a literature review. The five steps of the approach establish a defined set of guidelines for the researchers to follow, which may help ensure that their review is rigorous and comprehensive. Furthermore, using inclusion and exclusion criteria in stage three helps ensure that only high-quality studies are included in the review, which may help reduce bias and boost the dependability of the results. Overall, the technique provides a well-established and rigorous procedure for conducting a literature review. The updated approach adheres to the concepts stated by Saunders et al. (2019), which state that a literature review comprises the five steps listed below:

1. Formulating research questions

2. Searching for relevant literature from online and other sources

3. Selecting and evaluating studies based on criteria of inclusion and exclusion

4. Analysis and synthesis

5. Reporting the results

As a first step research agenda was set by defining the research questions which are as follows:

RQ1. How research on social entrepreneurship (SE), non-governmental organizations (NGOs), and globalization has changed over time?

RQ2. What are the most cited and influential authors in studies converging on the topics of social entrepreneurship (SE), NGOs, and globalization?

RQ3. What is the most influential work on the convergence of social entrepreneurship (SE), non-governmental organizations (NGOs), and globalization?

RQ4. What are the trends of cooperation among authors and countries in studies converging on the subjects of social entrepreneurship (SE), non-governmental organizations (NGOs), and globalization?

RQ5. What are the common themes of research and the most-referred theories among researchers in the field of studies converging on the issues of social entrepreneurship (SE), non-governmental organizations (NGOs), and globalization?

To achieve a wide-ranging overview of the convergence of dominant themes related to social entrepreneurship, NGOs, and globalization, a comprehensive literature review was conducted using five significant online databases: Scopus, Web of Science, JSTOR, ProQuest, and Science Direct. The search terms used in this study were "social entrepreneurship", "non-governmental", or "NGO", and "globalization". These terms had to be present in the title, abstract, or keywords of the studies. The search was limited to studies published between 2000 and 2022 and studies in English or with some English language information available. The results were further filtered based on the research area selected. Despite the filters, some irrelevant publications were still included. However, we were able to eliminate these irrelevant studies by reviewing their abstracts. Authors worked collectively on a single record file and applied inclusion

exclusion criteria which is delineated in Fig 1. A thematic literature review strategy was adopted, and 73 research papers were selected. Firstly, the titles, keywords, abstract summary, and outcome measured were tabulated using MS Excel, and key themes were identified using Voyant tools. Keyword search is an appropriate way to find the article relevant for the purpose of the study [42]. In the second stage, we performed bibliometric analysis of the literature study that elaborated upon SE or NGOs and globalization to help understand the intellectual structure of the research field. For our current study, we have used VOS Viewer because of its ability to display larger bibliometric maps in an easy-to-interpret way, which was not possible with the previous bibliometric software. Moreover, the software offers the added functionality of zooming, scrolling, and searching, which largely facilitates the detailed examination of a map.

This study conducted an exhaustive review of empirical research published in peer-reviewed journals between 2000 and 2021, mostly using the Web of Science (WOS) database. The researcher diligently preserved the acquired data. The retrieved data were imported as. CSV and. RIS files into the Refworks library for further analysis. In addition, a copy of each chosen article was preserved in the WOS saved list folder for future bibliometric analysis. The following section presents and analyzes bibliographic data from the WOS database using MS Excel, Voyant tools, Research Rabbit, and the VOS viewer. In research, triangulation refers to the use of various methods, data sources, or perspectives to answer a research issue. By combining evidence from numerous sources, this technique can improve the validity and reliability of research findings. Using various tools and using triangulation can help reduce biases. By merging many perspectives and data sources, it can also provide a more thorough and nuanced understanding of a phenomenon. Finally, triangulation can increase trust in research findings and the study's overall credibility. This research utilized multiple data analysis tools, such as the VOS viewer, Voyant tools, and Research Rabbit, to incorporate an element of triangulation to ensure that results were consistent across platforms.

## Findings and discussion

RQ1 was included in the research to better understand the development of SE, NGO, and globalization notions. Over the last decade, the focus has shifted away from outdated models and toward more sustainable ones [32]. RQ1 on the development of research literature production dynamics through time is addressed in Figs 2 and 3. 173 academics produced a total of 73 papers between 1999 and 2022; Fig 2 depicts the study on SE or NGOs and globalization by publication year. There has been a notable increase in this area of research in recent years, with about a third of the papers produced after 2015. While globalization has increased the scope and impact of non-governmental groups, it has also presented new challenges, such as increased competition for limited resources and lower visibility in an increasingly congested worldwide marketplace [37]. Recent study indicates a greater emphasis on various aspects of globalization, as well as the adoption of a sustainability component within NGOs and SE research.

Before identifying common themes, it is essential to understand the field of re-search on which a study has been centered in order to properly categorize it. The re-search articles featured in the paper are from 44 peer-reviewed journals in the fields of business and management, computer science, engineering, energy, the environment, psychology, the arts, and the humanities, among others. Meso Radar Chart of Citations, shown in Fig 3, reveals that the fields of management and economics account for more than seventy-five percent of the total number of citations. This is not surprising given that management studies and economics have paid more attention to SE, NGO, and globalization in recent decades.

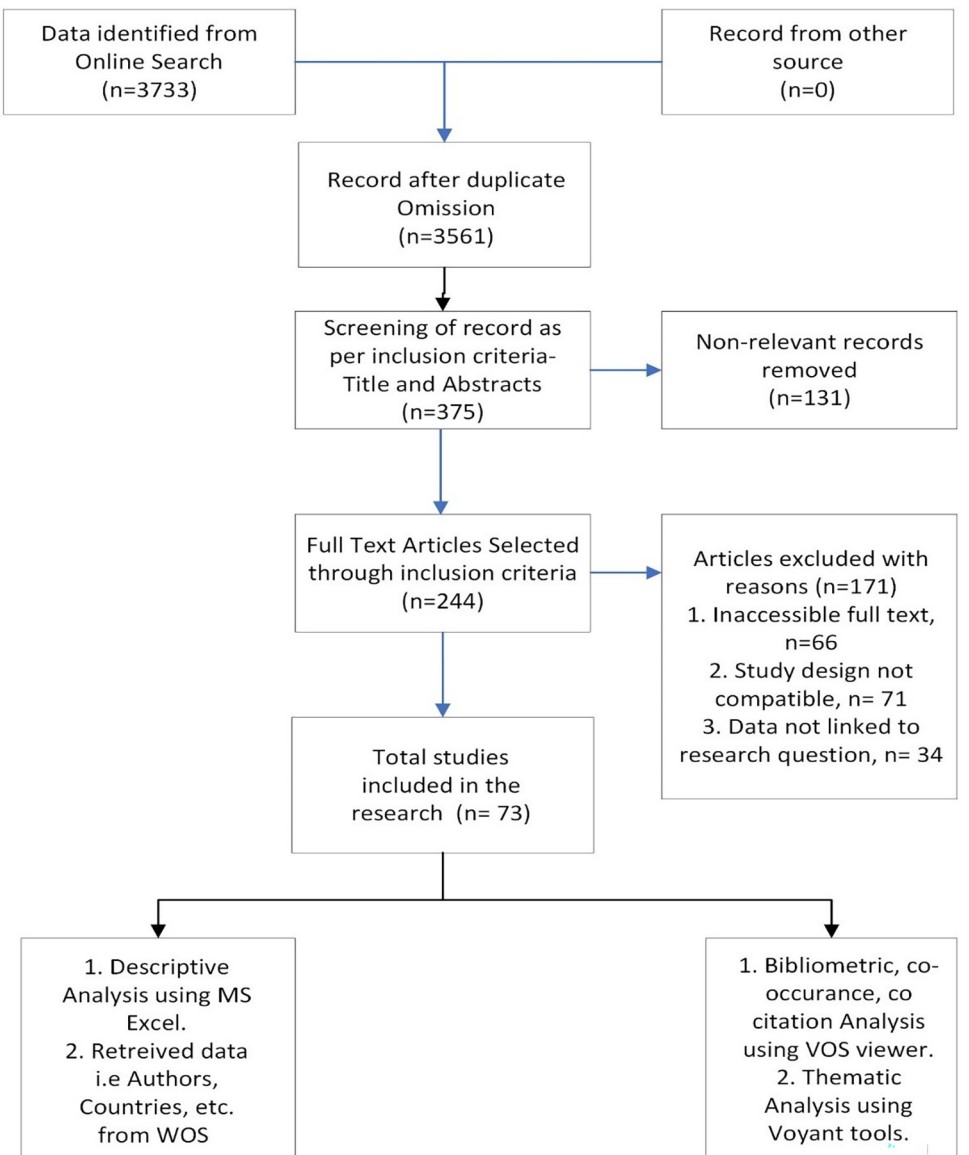

**Fig 1. PRISMA flow diagram for inclusion of studies and exclusion reasons, formulated by author using MS Visio (2016).**

Analyzing patterns of research cooperation is essential for a variety of reasons. First and foremost, studying collaboration patterns can aid academics in comprehending the progress of research activity. These statistics aid researchers in comprehending the topic and finding potential areas for future investigation. Second, cooperation patterns can help researchers comprehend the characteristics that drive collaboration in their field. Researchers may investigate whether researchers with comparable backgrounds collaborate more or whether certain research endeavors are more collaborative. Thirdly, studying cooperation trends can demonstrate how collaboration influences research quality and output. Researchers may examine whether collaborative research projects produce better findings or more influential articles than non-collaborative research initiatives. In order to respond to RQ1 and RQ4, an examination of cooperation indicated that 173 authors had co-authored research articles,

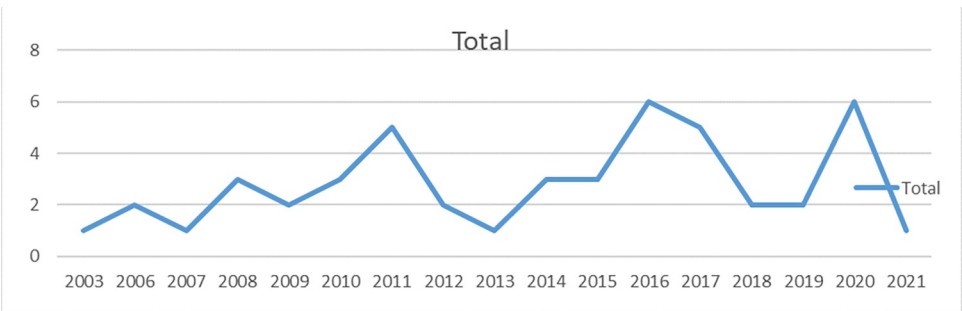

**Fig 2. Studies identified in our study, by year of publication, source: Formulated by author using MS Excel (2016).**

demonstrating an increase in author collaboration. 18 of the total 73 articles are research papers produced by a single author. 2.62 is the author cooperation index. 0.40 is the number of documents per author, whereas 2.50 is the value of authors per document. Specifically, MS Excel was used for bibliographic analysis [34]. Fig 2 depicts the number of articles every year that converge on SE, NGOs, and globalization. The Nonprofit and Voluntary Sector Quarterly [35] published one of the initial research articles outlining the effects of globalization on the NGO sector in 1999. This article begins by summarizing aspects of the NGO sector, including its rapid growth, new legitimacy and reputation, and national and regional variety. As globalization transforms the political economy in which NGOs operate, the conflicts, trade-offs, trends, and tactics offered by globalization are then examined. These descriptive analyses provide essential insights for addressing RQ4, i.e., the patterns of author and country collaboration in research.

To answer the second research question, we refer to Table 1 and its analysis for the most cited and influential authors in research that overlaps the topics of SE, NGOs, and globalization. Significant details about the prolific author's contribution and influence in the field of research were revealed by the author's analysis. In addition, it was deemed essential to identify the widely cited research articles that had opened up new avenues for future study. Regardless of when the article was published, the total number of citations per year demonstrates the article's influence. To identify the most influential articles published in the fields of SE, NGO, and globalization, we required a minimum of 100 citations.

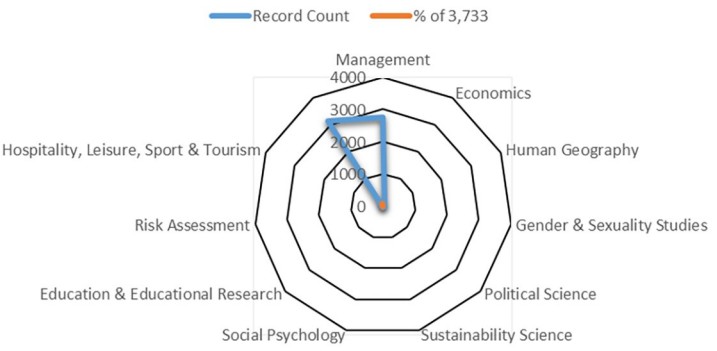

**Fig 3. Radar Chart of Citation topic Meso, source: Formulated by author using MS Excel (2016).**

**Table 1. Highly cited research papers converging on SE, NGO and globalization topics, source: Formulated by author using MS Excel (2016).**

| S. No. | Paper title | Authors | Year | Citations | C/Y |
|---|---|---|---|---|---|
| 1 | Social and Commercial Entrepreneurship: Same, Different, or Both? | J. Austin, H. Stevenson, Jane Wei-Skillern [43] | 2006 | 2959 | 184.94 |
| 2 | Social Entrepreneurship: Creating New Business Models to Serve the Poor | Christian Seelos, Johanna Mair [44] | 2005 | 954 | 56.12 |
| 3 | Corporate Social Responsibility, Public Policy, and Ngo Activism in Europe and the United States: An Institutional-Stakeholder Perspective. | J. Doh, T. Guay [45] | 2006 | 946 | 59.13 |
| 4 | Social Entrepreneurship: New Models of Sustainable Social Change | A. Nicholls [46] | 2008 | 921 | 65.79 |
| 5 | Social entrepreneurship: towards conceptualization | G. S. Mort, J. Weerawardena, K. Carnegie [47] | 2003 | 731 | 38.47 |
| 6 | Corporate social responsibility in the multinational enterprise: strategic and institutional approaches | Bryan W. Husted, David Bruce Allen [48] | 2006 | 687 | 42.94 |
| 7 | Globalization, Corporate Social Responsibility and poverty | R. Jenkins [49] | 2005 | 563 | 33.12 |
| 8 | Corporate Social Responsibility and Corporate Sustainability | Ivan Montiel [50] | 2008 | 537 | 38.36 |
| 9 | Social entrepreneurship: Key issues and concepts | S. Trevis Certo, Toyah L. Miller [51] | 2008 | 453 | 32.36 |
| 10 | Social Innovation and Social Entrepreneurship | W. Phillips, Hazel. Lee, A.Ghobadian, Nicholas O'Regan, P. James [52] | 2015 | 328 | 46.86 |
| 11 | Globalization and Commitment in Corporate Social Responsibility | Alwyn Lim, Kiyoteru Tsutsui [53] | 2012 | 311 | 31.10 |
| 12 | Corporate Social Responsibility | Philippe ugler, Jacylyn Y. J. Shi [54] | 2009 | 294 | 22.62 |
| 13 | Globalization and Corporate Social Responsibility | A. Scherer, G. Palazzo [55] | 2008 | 280 | 20.00 |
| 14 | Corporate social responsibility: The centerpiece of competing and complementary frameworks | Archie B. Carroll [56] | 2015 | 274 | 39.14 |
| 15 | Social entrepreneurship: A critical review of the concept | A. M. Peredo and M. McLean [57] | 2006 | 267 | 26.70 |
| 16 | From Management Systems to Corporate Social Responsibility | Gerard I.J.M. Zwetsloot [58] | 2003 | 196 | 10.32 |
| 17 | Corporate Social Responsibility (CSR) and Innovation–The Drivers of Business Growth? | Gadaf Rexhepi, Selma Kurtishi, Gjilnaipe Bexheti [59] | 2013 | 131 | 14.56 |
| 18 | Globalization and Corporate Social Responsibility: How Non-Governmental Organizations Influence Labor and Environmental Codes of Conduct | Jonathan P. Doh, Terrence R. Guay [45] | 2006 | 117 | 6.50 |

Table 1 and its analysis provide essential insight into the most influential work in terms of citations and impact on following research work, so answering the second research question RQ2, "Who are the most referenced and influential writers in SE, NGO, and globalization?" Table 1 reveals that the most cited work is by Austin et al. (2006).

This article has been cited 2,959 times since its publication in 2006, demonstrating its significance [37]. This article compares commercial and social entrepreneurship using a well-known method for studying business entrepreneurship. The study reveals significant similarities and differences between these two types of entrepreneurship and provides a framework for approaching the social entrepreneurship process in a more systematic and effective manner. This study examines the implications of research on social entrepreneurship for practitioners and academics. It is important to analyze trends in researcher collaboration because it can provide valuable insights into the state of the field, the factors that facilitate cooperation, and the effect of collaboration on research quality and productivity [58]. "Social Entrepreneurship: Creating New Business Models to Serve the Poor" is the second most cited work, having been cited 954 times since its publication. The article examines how to scale innovative, inclusive growth solutions to combat poverty. Using a realist perspective, the authors analyze an organization that has achieved sustained success in this area for over 20 years. They propose "organizational closure" as a key competence for achieving scale in poverty contexts and suggest implications for practice and research.

The third most cited research is "Corporate Social Responsibility, Public Policy, and NGO Activism in Europe and the United States: An Institutional-Stakeholder Perspective." This

paper examines how institutional differences between Europe and the United States affect corporate social responsibility (CSR) expectations. Using case studies on issues such as global warming and access to pharmaceuticals, the authors find that varying institutional structures and political legacies affect how CSR is prioritized and implemented. The fourth study, "Social Entrepreneurship: New Models of Sustainable Social Change," covers the activities of various groups, such as activists, NGOs, and corporations, who address social issues through innovative methods. The study includes contributions from academics, policymakers, and practitioners, with an emphasis on the development of social impact models that are sustainable. The objective is to define the term "social entrepreneurship," highlight relevant projects, and develop conceptual frameworks.

Social entrepreneurship: towards conceptualization, the fifth most cited work, emphasizes the need to study social entrepreneurship, a concept involving the establishment and innovation of social enterprises. The authors propose a multidimensional model of social entrepreneurship that emphasizes virtuous behavior, unity of purpose and action, recognition of opportunities to create social value, and key decision-making traits. The sixth most-cited article, "Corporate Social Responsibility in the Multinational Enterprise: Strategic and Institutional Approaches," examines the connection between global and local corporate social responsibility (CSR) and international organizational strategy. Using a survey of multinational firms operating in Mexico, the authors conclude that institutional pressures rather than strategic analysis drive CSR-related decision-making, and they provide implications for management, research, and public policy.

The seventh most-cited study is titled "Globalization, Corporate Social Responsibility, and Poverty." The article discusses how corporate social responsibility (CSR) has become a popular topic among development practitioners, despite criticism from development NGOs and support from official development agencies. The article analyzes the connections between foreign direct investment and poverty, using a framework to illustrate the limitations of CSR's impact on poverty reduction and concluding that CSR is unlikely to play as significant a role in reducing poverty in developing countries as its proponent's assert. The eighth most-cited study is titled "Corporate Social Responsibility and Corporate Sustainability." This article examines the evolution of the definitions of corporate social responsibility (CSR) and corporate sustainability (CS) in the management literature. The author concludes that there is no clear distinction between the two terms and proposes that scholars who study these issues collaborate more to reshape the field.

The ninth most cited work, "Social entrepreneurship: Key issues and Concepts," discusses the various business models within the fair trade system and the need for a greater mainstream understanding of fair trade. The author argues that fair trade needs to address both the value chain and institutional governance to be more than just another consumer standard and suggests policy implications for fair trade institutions. "Social Innovation and Social Entrepreneurship" is the tenth most cited piece of literature. The article discusses how corporate social responsibility (CSR) initiatives can improve the performance of for-profit businesses in ways that go beyond satisfying the interests of stakeholders. The authors review research on social entrepreneurship and social innovation and synthesize it into an analytical framework for future studies of social innovation and social entrepreneurship using a "systems of innovation" approach.

Figs 4 and 5 and their related analyses are provided to elaborate on RQ3, which aims to explain patterns of collaboration between authors and nations in SE, NGO, and globalization research. The co-authorship network displays the extent of collaboration across scholars, organizations, and countries. Collaboration has a synergistic effect [42]. Furthermore, collaborative research produces more innovative scientific output and higher-quality research papers. In

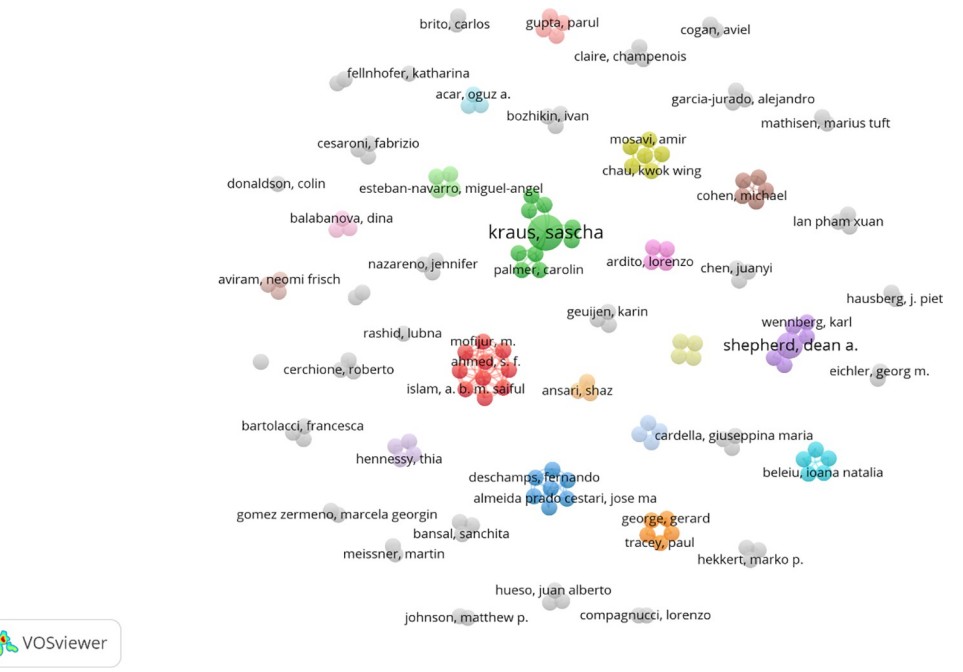

**Fig 4. Co-authorship network of authors selected papers, source: Formulated by author using VOS viewer.**

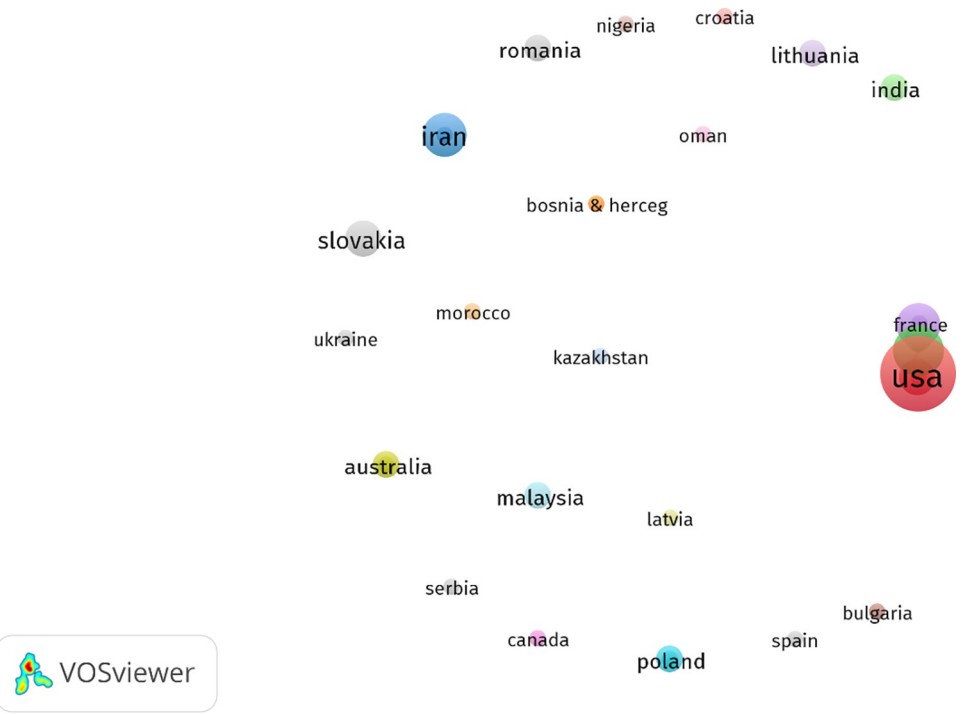

**Fig 5. Co-authorship network of authors grouped by affiliated countries, source: Formulated by author using VOS viewer.**

general, researchers interact and contribute collectively to the production of scientific papers, which results in a greater number and quality of scientific output due to the involvement of each individual. The analysis was carried out using VOS Viewer. Fig 4 depicts the co-authorship network of authors who have co-authored at least one scientific article. To qualify for the requirements, experts must have coauthored at least one document and been cited at least 50 times between 2000 and 2020.

Study by Wardil & Hauert (2015) has identified that cooperation patterns reveal areas of high collaboration as well as areas of low collaboration [60]. This data can assist researchers in identifying collaboration opportunities and minimizing duplication. Second, collaborative patterns may offer an overview of a topic's present position. For example, if researchers on a certain problem effectively collaborate, it may signal that the area is mature and well-established, with a robust research background. Insufficient collaboration among researchers, on the other hand, may indicate that the issue is still growing or that there are gaps in the study. Third, patterns of collaboration may suggest research possibilities. For example, if researchers are heavily involved in a sub-topic within a subject, it may indicate that there is a lot of interest in it and that additional research in this area would be useful. Overall, assessing the patterns of academic cooperation on a certain problem may provide significant insight into the state of the field and help identify opportunities for collaboration, future research routes, and prospective research gaps [42]. The review of Fig 4 resulted in the grouping of 173 writers into 47 clusters, with each cluster represented by a distinct color. Fernandes Christina I, J. Halberstadt, Kraus Sascha, Palmer Carolin, Kailer Norbert, Kallinger Friedrich Lukas, Spitzer Jonathan, Vallaster Christine, Lindahl, Jose M. Merigo, and Nielsen Annika, shown as a green cluster, have co-authored three documents with nine shared links. The second most prevalent The coauthor ship network of author-affiliated nations with at least one publication between 2000 and 2020 is depicted in Fig 4. A minimum of 10 citations per nation was used to determine the threshold limitations, resulting in a total of 31 countries. The United States emerged as having the strongest total connection strength among all nations. There are no connections between the nations of Croatia, Lithuania, Kazakhstan, Nigeria, Ukraine, Iran, Romania, Oman, and Malaysia. On the other hand, there exists a robust collaboration network between nations like the United States, France, Italy, and Australia. The United States emerged as having the strongest total connection strength among all nations.

An author keyword analysis was performed to capture the prevalent themes and transfer of ideas among scholars. As a result, we did an author keyword analysis to better understand the previous research trend in SE, NGOs, and globalization. We constructed a network of keyword co-occurrences using VOS viewer version 1.6.18. In the beginning, 103 keywords were retrieved from a list of 73 papers. The keywords were filtered to a minimum of two occurrences in order to generate the co-occurrence network of the most frequently used author keywords, yielding a total of 26 keywords. The minimal threshold criterion was met by 26 of the 103 keywords. The co-occurrence network of commonly used author terms is depicted in Fig 7.

This section examines the thematic organization of the included SE, NGO, and globalization-related publications. The X-axis depicts centrality (i.e., the degree of interaction of a network cluster with other clusters) and indicates the significance of a theme in Fig 6. The Y-axis represents density, or the internal strength of a cluster network, and can be interpreted as a measure of the development of the theme. Consequently, the first quadrant contains motor-related themes. This quadrant contains well-developed themes, such as Entrepreneurship and Globalization; the second quadrant contains highly developed and isolated themes, such as Social entrepreneurship; and the third quadrant contains emerging or declining themes. The analysis reveals that CSR-related topics receive substantial investigation, but this trend is

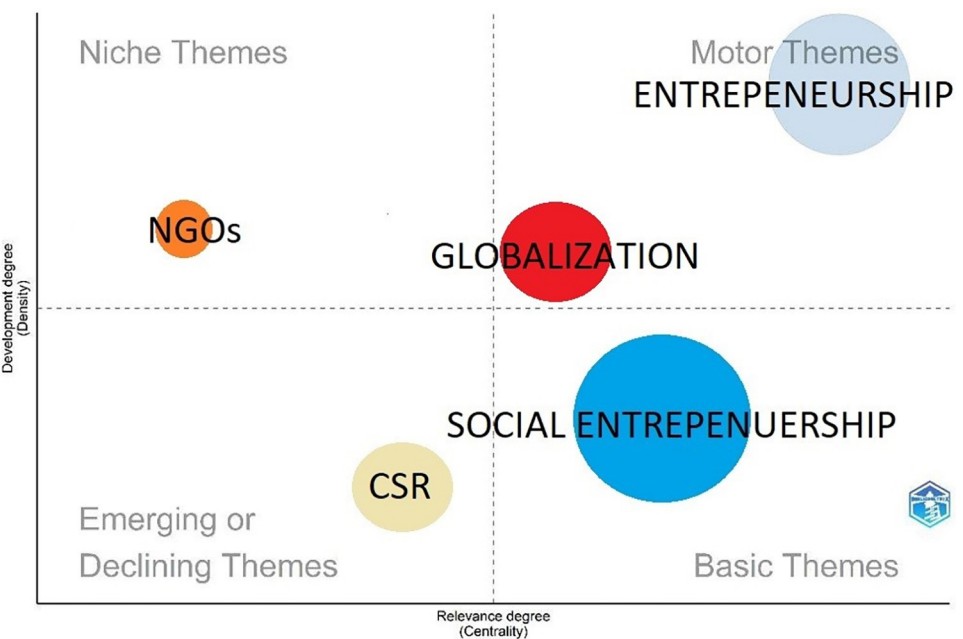

**Fig 6. Thematic map of studies showing dominant themes, source: Formulated by author.**

waning as organizations adopt more sustainable practices. NGO-related items are located in the fourth quadrant, which is considered a Niche.

The analysis of Figs 6 and 7 indicates that "globalization" and "social enterprise" have received the most attention, respectively. Innovation, social globalization, inclusive growth, and internationalization have emerged as some of the most frequently employed terms by

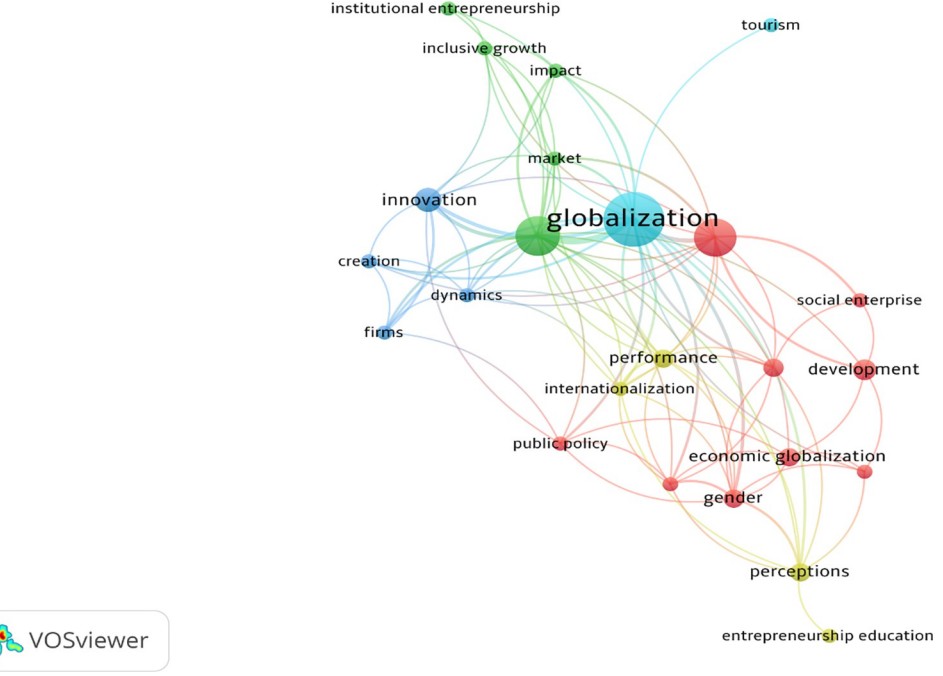

**Fig 7. Co-occurrence of author's specific keyword, source: Formulated by author using VOS viewer.**

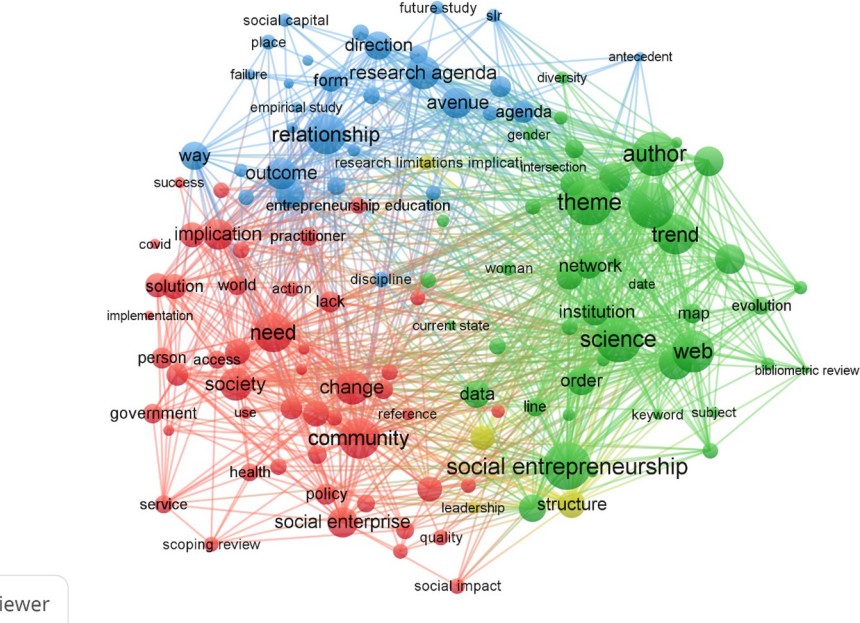

**Fig 8. Identifying constructs in studies on NGOs, social enterprises, and globalization using cluster analysis, source: Formulated by author using VOS viewer.**

authors. The research yielded a total of 26 keywords, which were organized into six clusters. Each cluster was colored differently, and the total link strength was 139.

By looking at the themes that emerge across multiple studies, researchers can gain a better sense of the underlying factors that are driving research in a particular field, and can identify any gaps or inconsistencies in the existing research. Figs 7 and 9 are provided to address RQ5, which is to highlight the common topics of study and the most-referred themes among researchers in the area of studies converging on SE, NGOs, and globalization. The author's keyword analysis gives useful information. Ac-cording to a review of the techniques employed by highly cited studies, traditional theoretical adoption models, such as the globalization index, sustainability theory, and entrepreneurism theory, are commonly used by academics to develop theoretical models. Researchers have used these models to investigate the impact of globalization on SE and NGOs. Figs 7 and 8 show that researchers frequently utilize various themes interchangeably, and some studies have even used hybrid ideas to more comprehensively explore the effect of globalization. Second, when analyzed in conjunction with Fig 5, it is clear that research is primarily focused on developed countries such as the United States, the European Union, and western nations because data on globalization, social enterprise, and nongovernmental organizations (NGO) are more readily available for these economies than for less developed nations. Fig 9 shows a word cloud for frequent topics in the chosen study. The word cloud was constructed by first summarizing the research' titles, key findings, and abstract summaries, and then evaluating the data using analytical tools. The key themes are represented by words of various colors, and the size of each phrase represents its frequency of recurrence. Entrepreneurship, social emphasis, sustainability, and innovation remain important re-search interests.

Using cluster analysis, Fig 8 identifies the common themes in studies on NGOs, social enterprises, and globalization, represented by four clusters. Analyzing the clusters reveals that researchers have focused on social entrepreneurship from various perspectives using methods such as thematic analysis and network analysis, exploring the impact of variables such as

**Fig 9. World cloud of recurring themes, source: Formulated by author using Voyant tools.**

government, society, and community. They have also examined management practices, leadership roles, and organizational structures. Overall, the numerous links and high density of research suggest that the topic has been extensively researched from various constructs and perspectives.

Fig 10 of keywords for the period 2000–2022 reveals four related circles: (I) Corporate social responsibility and Globalization, (ii) SE and Sustainability, (iii) NGOs and social change, and (iv) CSR and business model (Fig 7). Many studies have examined the impact of globalization on CSR. The concept of NGOs for social change has evolved into more sustainable models, such as those proposed by social enterprises. While CSR as a business model has received significant attention in previous decades, the focus is now shifting towards more sustainable models, such as those proposed by SE. Given the disruptions caused by the COVID-19 pandemic, it is reasonable to assume that research on the effects of globalization on SE and NGOs is gaining traction and will be effectively utilized. In recent years, the nonprofit sector has made substantial contributions to the fields of SE and sustainable practices.

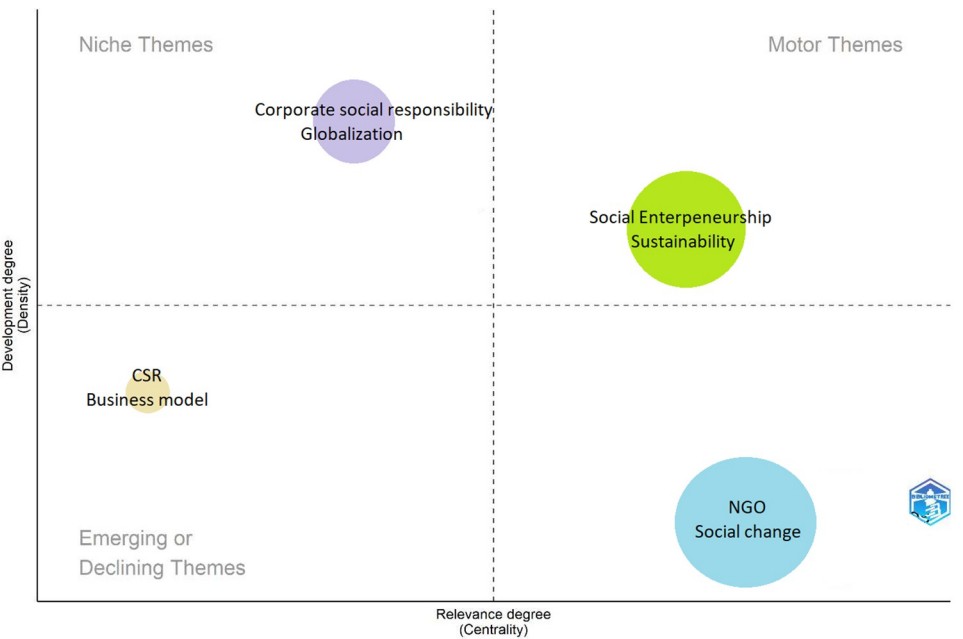

**Fig 10. Thematic map showing various dimensions of topics (2000–2022).** Source: Created by the authors.

Since 2013, there has been a significant increase in the amount of research published in reputable peer-reviewed journals. An analysis of scientific output by country reveals that established economies in the West, such as the United States, the United Kingdom, and the European Union, conduct the most research. According to this study, the majority of research examines SE and NGOs from various perspectives, but relatively few studies have examined both SE and NGOs as well as the effect of globalization on mutual adaptation. The findings of this study indicate a need for additional research on the effects of globalization on SE and NGOs, particularly in terms of empirical and statistical data collection.

## Conclusions

This study is important because it provides a comprehensive review of the literature on social entrepreneurship (SE), non-governmental organizations (NGOs), and globalization, all of which are critical topics for understanding how to effect long-term social change in the changing economic and social landscape shaped by globalization. To achieve these aims, we conducted a thematic and bibliometric analysis of the relevant research. This required collecting and categorizing peer-reviewed papers from databases such as Scopus, Web of Science, and Science Direct, in addition to supplementing the search with existing databases and bibliographies. Using this systematic review method, we were able to identify key themes and patterns in the literature, which contributed to our understanding of these complex and interrelated topics. Researchers from a variety of fields and institutions are publishing their findings in prestigious peer-reviewed journals, indicating that SE research is on the rise. A country-by-country analysis, on the other hand, reveals that although US research papers are frequently cited, they may not always reflect a significant trend of development over time. The findings are helpful for practitioners, policymakers, and academics interested in SE, NGOs, and globalization and can guide future research and practice in these fields. This study contributes to the understanding of the relationship between globalization, entrepreneurship, and social and environmental concerns. It supports the conclusion that globalization has increased awareness

of the role of entrepreneurship in addressing social problems and contributing to the creation of wealth. The findings of this research support previous studies that have identified globalization as a key factor in increasing awareness of the role of entrepreneurship in wealth creation and addressing social issues [59]. Globalization has led to the liberalization of national economies, demographic shifts, institutional and political failures, and technological improvements, all of which have contributed to a greater social consciousness and a desire for firms to engage in social projects [60]. Globalization has also led to the development of social entrepreneurship and the growth of NGOs as important players in addressing social and environmental challenges. Understanding these trends and their implications is important for practitioners, policymakers, and researchers. These trends have encouraged the development of social entrepreneurship and the growth of NGOs as important players in addressing social and environmental challenges.

This study examined peer-reviewed literature to discover the dominance of qualitative research methodologies in the fields of social entrepreneurship (SE), non-governmental organizations (NGOs), and globalization. We also examined the network of co-authorship or scientific collaboration to see if there were any trends in research convergence among particular writers and nations. Our research revealed that there is not always a correlation between writers and countries, especially in underdeveloped nations. We conducted a variety of bibliometric analyses using the VOS viewer. Our findings reveal a paucity of quantitative research in the fields of SE, NGOs, and globalization, with scholars focusing instead on conceptualization and theoretical methods. This may be because collecting data from SEs and NGOs is more difficult than collecting conceptual or theoretical data, particularly in developing nations. This study emphasizes the need for additional research on these topics, particularly research employing a variety of methods and approaches. This study is important because it demonstrates the predominance of qualitative research methods in the fields of social entrepreneurship (SE), non-governmental organizations (NGOs), and globalization. It also reveals a trend toward research convergence among certain authors and nations, as well as the absence of inextricable links between authors and countries, especially in developing nations. Utilizing the VOS viewer as a bibliometric tool enables a comprehensive examination of the relevant literature. Our findings indicate the need for additional research on SE, NGOs, and globalization, particularly research employing a variety of methodologies and approaches. SE, NGOs, and globalization are all intricate and intertwined topics with serious implications for practitioners, policymakers, and academics. Understanding the changing nature of non-governmental organizations (NGOs) and social concerns, as well as the effect of globalization on these issues, is essential for designing effective management strategies and addressing social and environmental problems. In order to increase our understanding and contribute to positive social change, it is essential that we continue to engage in research on these topics.

In future research, a variety of bibliometric tools, such as BibExcel, HistCite, and Gephi, can be utilized to improve visualization and provide a more comprehensive evaluation. Scientific mapping analysis software, such as Cite Space, CiteNetExplore, Sci2Tool, and SciMat, can also be used to characterize the relationship between multiple units of analysis. However, it should be noted that SE, NGOs, and globalization are all expansive, context-dependent concepts with numerous facets, making it difficult to draw broad conclusions about these issues. The evaluation of the included papers demonstrates that the current understanding of charity has shifted from CSR and NGO-based models to one that is more sustainable, as suggested by SE. Non-governmental organizations (NGOs) are not established with the intent of conducting business and rely on government and donor funding [13]. In addition, they may encounter capacity issues such as access to high-quality human resources, training resources, and network development [35]. In socially oriented organizations, the concept of sustainability should

not be ignored, but in developing nations, it would require government and international and local donor agency interventions to achieve widespread recognition. This would not be possible without the participation and growth of existing non-governmental organizations. Entrepreneurial initiatives, particularly those with a social mission, have a bright and promising future because they produce lasting results. Future research should investigate page rank analysis to differentiate between the popularity and prestige of a research paper. This would allow for a more thorough examination of the relevance of the cited papers and assist in identifying the most significant research paper in the field of study. This study provides a detailed overview of research on SE, NGOs, and globalization that will be particularly useful for academics and practitioners, despite the limitations mentioned above. This will assist in advancing and expanding research in this field. It is essential for policymakers to comprehend the impact of globalization on social entrepreneurship and nongovernmental organizations (NGOs) in order to develop effective policies that encourage social entrepreneurship while addressing the broader social and economic consequences of globalization. By doing so, policymakers can ensure that globalization benefits society as a whole and encourages social entrepreneurship as a means of addressing global issues. This study has contributed to the existing body of knowledge by highlighting highly cited papers that can be used to identify key themes and concepts regarding SE, NGOs, and the impact of globalization. This may include providing training and support to social entrepreneurs entering new markets, assisting them in obtaining funds and other resources, and implementing policies to encourage and promote social entrepreneurship. It also assists policymakers in comprehending the opportunities and obstacles presented by globalization to social entrepreneurs.

## Supporting information

**S1 Checklist. PRISMA 2020 checklist.**
(DOCX)

## Author Contributions

**Conceptualization:** Muhammad Rizwan Hussain, Khalid Bin Muhammad.

**Data curation:** György Norbert Szabados, Shah Ali Murtaza.

**Formal analysis:** Khalid Bin Muhammad.

**Investigation:** Shah Ali Murtaza.

**Methodology:** Muhammad Rizwan Hussain.

**Project administration:** György Norbert Szabados, Sevinj Omarli, Edina Molnár.

**Supervision:** Edina Molnár.

**Validation:** Sevinj Omarli.

**Writing – review & editing:** Muhammad Rizwan Hussain, Sevinj Omarli, Shah Ali Murtaza.

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
