## [Decision Letter · Decision Letter 0]

16 Feb 2023

PONE-D-23-01240Examining the convergence of dominant themes related to Social Entrepreneurship, NGOs and Globalization – A

Systematic Literature ReviewPLOS ONE

Dear Dr. Murtaza,

Thank you for submitting your manuscript to PLOS ONE. After careful consideration, we feel that it has merit but does not fully meet PLOS ONE’s publication criteria as it currently stands. Therefore, we invite you to submit a revised version of the manuscript that addresses the points raised during the review process.

We look forward to receiving your revised manuscript.

Kind regards,

Muhammad Farhan Bashir

Academic Editor

PLOS ONE

Journal Requirements:

"No"

"No"

4. Please ensure that you refer to Figure 1 in your text as, if accepted, production will need this reference to link the reader to the figure.

Reviewers' comments:

Reviewer's Responses to Questions

**Comments to the Author**

1. Is the manuscript technically sound, and do the data support the conclusions?

Reviewer #1: Yes

Reviewer #2: Partly

2. Has the statistical analysis been performed appropriately and rigorously? 

Reviewer #1: No

Reviewer #2: No

3. Have the authors made all data underlying the findings in their manuscript fully available?

Reviewer #1: Yes

Reviewer #2: No

4. Is the manuscript presented in an intelligible fashion and written in standard English?

Reviewer #1: Yes

Reviewer #2: No

5. Review Comments to the Author

Reviewer #1: Comments on Manuscript “Examining the convergence of dominant themes related to Social Entrepreneurship, NGOs and Globalization – A Systematic Literature Review” Thank you very much for giving me the opportunity to review the manuscript. I must appreciate the work authors have performed. I have reviewed this paper thoroughly and a few suggestions are given below:

Introduction:

I would suggest improving this section by making it more explainable in terms of the Convergence of dominant with the significant Globalization included in the study.

The last statement in the introduction is redundant; it would be more appropriate to describe the study's implications here.

Regarding the introduction, more examples of the significance of social economy and the evolving role of NGOs should be included.

Avoid using abbreviations (explain it in detail in the first stance and then use the acronyms.)

Literature Review:

This section is well-aligned and presents an effective blend of recent and past studies. Overall, this section is appropriate in my opinion. Kindly add some studies in literature review section;

https://doi.org/10.3390/su14031054

https://doi.org/10.1007/s11356-021-17438-x

https://doi.org/10.1108/LHT-03-2021-0113

https://doi.org/10.1007/s11356-022-19718-6

https://doi.org/10.1007/s11356-022-19628-7

https://doi.org/10.1007/s11356-022-19954-w

https://doi.org/10.1007/s11356-022-20178-1

https://doi.org/10.1007/s11356-022-20922-7

Methodology:

This section is well-written and well-explained. Referring to my comments in the Introduction section, try to explain each abbreviation at first and then use its short form.

Empirical Results:

Results are explained in detail. Must add much more explanations and interpretations for the results, which are not enough. It is suggested to compare the results of the present research with some similar studies which is done before (more justification is needed).

Conclusion:

Please make sure your conclusions section underscores the scientific value-added of your paper and the applicability of your findings/results, as indicated previously. Please revise your conclusion part into more detail. It would be best if you enhanced your contributions, and limitations, underscore the scientific value-added of your paper, and the applicability of your findings/results and future study in this session.

I hope these comments would enhance the quality of the manuscript to make it an appropriate fit for the Journal of PLOS ONE Readership.

Reviewer #2: 1. only one highly cited research paper has been explained whereas there are many papers with high citations on the topic in table 1 please include at least 10 articles

2. In figure 5 China is not present in it which seems very strange to me because this is not possible without Chinese collaboration

3. It seems that article selection process is not correct, enlist keywords which you have searched and then the query searched on Scopus and WOS

4. There could be cluster analysis at VOSviewer which could systematically explain constructs in the field.

5. On biblioshiney there is a process of theme identification which could be used to explain different dimensions of topic here.

6. PLOS authors have the option to publish the peer review history of their article (what does this mean?). If published, this will include your full peer review and any attached files.

Reviewer #1: **Yes: **KASHIF ABBASS

Reviewer #2: **Yes: **Hafiz Muhammad Arslan

---

## [Author Response · Author response to Decision Letter 0]

22 Feb 2023

Dear Editor and Reviewers, The revised version has been submitted and all recommended changes have been inculcated in the revised manuscript.

---

## [Decision Letter · Decision Letter 1]

2 Mar 2023

Examining the convergence of dominant themes related to Social Entrepreneurship, NGOs and Globalization – A Systematic Literature Review

PONE-D-23-01240R1

Dear Dr. Murtaza,

We’re pleased to inform you that your manuscript has been judged scientifically suitable for publication and will be formally accepted for publication once it meets all outstanding technical requirements.

Kind regards,

Muhammad Farhan Bashir

Academic Editor

PLOS ONE

Additional Editor Comments (optional):

Reviewers' comments:

Reviewer's Responses to Questions

**Comments to the Author**

1. If the authors have adequately addressed your comments raised in a previous round of review and you feel that this manuscript is now acceptable for publication, you may indicate that here to bypass the “Comments to the Author” section, enter your conflict of interest statement in the “Confidential to Editor” section, and submit your "Accept" recommendation.

Reviewer #1: All comments have been addressed

Reviewer #2: All comments have been addressed

2. Is the manuscript technically sound, and do the data support the conclusions?

Reviewer #1: Yes

Reviewer #2: Yes

3. Has the statistical analysis been performed appropriately and rigorously? 

Reviewer #1: Yes

Reviewer #2: N/A

4. Have the authors made all data underlying the findings in their manuscript fully available?

Reviewer #1: Yes

Reviewer #2: Yes

5. Is the manuscript presented in an intelligible fashion and written in standard English?

Reviewer #1: Yes

Reviewer #2: Yes

6. Review Comments to the Author

Reviewer #1: All comments has been addressed properly. So, I am happy to informed you that paper is accepted for publication

Reviewer #2: (No Response)

7. PLOS authors have the option to publish the peer review history of their article (what does this mean?). If published, this will include your full peer review and any attached files.

Reviewer #1: **Yes: **KASHIF ABBASS

Reviewer #2: **Yes: **Hafiz Muhammad Arslan

---

## [Editor Report · Acceptance letter]

7 Mar 2023

PONE-D-23-01240R1 

Examining the convergence of dominant themes related to Social Entrepreneurship, NGOs and Globalization – A Systematic Literature Review 

Dear Dr. Murtaza:

I'm pleased to inform you that your manuscript has been deemed suitable for publication in PLOS ONE. Congratulations! Your manuscript is now with our production department. 

Kind regards, 

on behalf of

Dr Muhammad Farhan Bashir 

Academic Editor

PLOS ONE